# Years of life lost to COVID-19 in 49 countries: A gender- and life cycle-based analysis of the first two years of the pandemic

Oscar Espinosa[1], Jeferson Ramos[1], Maylen Liseth Rojas-Botero[2], Julián Alfredo Fernández-Niño[3]*

1 Economic Models and Quantitative Methods Research Group, Centro de Investigaciones para el Desarrollo, Universidad Nacional de Colombia, Bogotá, D.C., Colombia, 2 Facultad Nacional de Salud Pública, Universidad de Antioquia, Medellín, Colombia, 3 Johns Hopkins Bloomberg School of Public Health, Johns Hopkins University, Baltimore, Maryland, United States of America

* jferna53@jhu.edu

**Data Availability Statement:** All the databases used for this analysis are public and can be directly accessed. The links for the databases are below: i)

## Abstract

Specific mortality rates have been widely used to monitor the main impacts of the COVID-19 pandemic; however, a more meaningful measure is the Years of Life Lost (YLL) due to the disease, considering it takes into account the premature nature of each death. We estimated the YLL due to COVID-19 between January 2020 and December 2021 in 49 countries for which information was available, developing an analytical method that mathematically refines that proposed by the World Health Organization. We then calculated YLL rates overall, as well as by sex and life cycle. Additionally, we estimated the national cost-effective budgets required to manage COVID-19 from a health system perspective. During the two years of analysis, we estimated that 85.6 million years of life were lost due to COVID-19 in the 49 countries studied. However, due to a lack of data, we were unable to analyze the burden of COVID-19 in about 75% of the countries in the world. We found no difference in the magnitude of YLL rates by gender but did find differences according to life cycle, with older adults contributing the greatest burden of YLL. The COVID-19 pandemic has posed a significant burden of disease, which has varied between countries. However, due to the lack of quality and disaggregated data, it has been difficult to monitor and compare the pandemic internationally. Therefore, it is imperative to strengthen health information systems in order to prepare for future pandemics as well as to evaluate their impacts.

## Introduction

The COVID-19 pandemic has had significant global repercussions. Since its identification, in 2019, through December 31, 2021, 288.8 million cases and 5.5 million deaths were registered worldwide [1]. To understand its impacts, multiple approaches need to be applied from various disciplines in specific populations and locations.

During the course of this social, economic and health crisis, SARS-CoV-2 infection cases and specific mortality rates have often been used to assess and compare the impacts of the

for the data on the age of death by COVID-19, sex and country: COVerAGE-DB, https://osf.io/mpwjq/; ii) for life expectancy by sex and without differentiation by sex: United Nations, https://population.un.org/wpp/Download/Standard/MostUsed/. For England, Scotland and the Island of Jersey, the information was taken from their official institutions: https://opendata.gov.je/dataset/2021-census/resource/96bc726b-d820-4be1-933a-7080b139fd24 and https://www.nomisweb.co.uk/datasets/ppsyoa; iii) for deaths from covid worldwide: https://ourworldindata.org/explorers/coronavirus-data-explorer; iv) for utilities ponderations (of the United Kingdom), according to the life cycle and sex: https://pubmed.ncbi.nlm.nih.gov/25692211/; v) for cost-effectiveness thresholds (CET) per country (conservative scenario): https://pubmed.ncbi.nlm.nih.gov/27987642/; and vi) for national consumer price indices: World Bank, https://databank.worldbank.org/data.

**Funding:** The authors received no specific funding for this work.

**Competing interests:** The authors have declared that no competing interests exist.

COVID-19 pandemic [2], as well as to evaluate the performance of governmental actions between regions and countries [3]. However, those measures have some limitations, insofar as they assign the same weight to all registered deaths, regardless of the age at which they occurred. This is an important limitation in a phenomenon such as COVID-19, where the risk of death differs according to age, with the highest death rates occurring among the elderly population [4].

Therefore, countries require more precise measures to better understand the pandemic's social and public health impacts. Estimating years of life lost (YLL) to COVID-19 is recommended as a more accurate approach to quantify at least the direct effects of the pandemic [5]. In particular, YLL takes into account the age at which each death occurs, indicating the loss faced by society as a result of premature deaths [6], in this case, due to COVID-19.

However, for YLL estimates to be accurate, countries must have robust health information systems that allow data to be disaggregated, at least by sex and age. Other crucial elements include testing capacity, effective access to health services, and the ability of national epidemiological surveillance to identify, categorize, and codify deaths from the SARS-CoV-2 infection. Perhaps as never before, this pandemic demonstrated the need for stronger information systems that make it possible to manage health data and generate evidence for action. Unfortunately, not all countries met these criteria, which has led to the underestimation of health impacts related to the pandemic at a societal level [7]

Several studies have calculated YLL due to COVID-19 with different populations and in various locations, some with a national scope [8–10] and others by comparing multiple countries [11–13]. In this study, we estimated YLL to COVID-19 in 49 countries worldwide, where health information systems met the overall requirements, by sex and age groups, and for the first two years of the pandemic by using an adaptation of the analytical method developed by the World Health Organization and proposed by Vieira et al. y Williams et al. [12–14]. Additionally, we explored the budget at which the national management of the COVID-19 pandemic would be cost-effective. This study provides a complementary approach to better understand and compare the direct effects of COVID-19 in terms of premature mortality in different countries around the world.

## Materials and methods

### Countries studied and sources of information

Data on the age of death by COVID-19, sex and country was obtained from microdata of COVerAGE-DB [15,16]. This is an open access database born from an international collaboration of more than 70 researchers from multiple disciplines, where cumulative counts of COVID-19 confirmed cases, deaths, tests and vaccines by age and sex are included in a centralized, standardized, age-harmonized and fully reproducible manner. COVerAGE-DB is considered a benchmark in the design and implementation of COVID-19 data processing and validation [15,16].

For this research, countries with complete data on COVID-19 mortality from January 2020 to December 2021 were considered. We found information on mortality by sex (from age zero) for both 2020 and 2021 in 36 countries, and without differentiation by sex in an additional 13 countries. In addition, we carried out a search in the official government institutions website of each of the more than 190 countries, effectively verifying that only 49 met the inclusion requirement to be included in this analysis. On the other hand, life expectancy by sex and without differentiation by sex was taken from the United Nations website [17] (for England, Scotland and the Island of Jersey, the information was taken from their official institutions).

The population by sex and single age, necessary to calculate YLL rates, was also used from this last source of information.

As a result, for Europe, we managed to extract information from 25 countries (Austria, Belgium, Bulgaria, Czechia, Denmark, England, France, Finland, Germany, Greece, Hungary, Island of Jersey, Italy, Moldova, Netherlands, Northern Ireland, Norway, Portugal, Scotland, Slovakia, Slovenia, Spain, Sweden, Switzerland, Ukraine), which represent 66.4% of its total population as a continent (in 2021). In the Americas there were 12 nations (Argentina, Brazil, Canada, Chile, Colombia, Haiti, Jamaica, Mexico, Paraguay, Peru, Uruguay, USA), which turns out to be 90.2% of its total population. In Oceania 2 countries (Australia, New Zealand) representing 69.8% of its total population, in Africa 2 nations (Somalia, Togo) that host 1.8% of the population of the entire continent, and in Asia 8 countries (Afghanistan, Bangladesh, Japan, Indonesia, Israel, Nepal, Philippines, South Korea), which represents 17.3% of its total population.

## Statistical analysis

To find the YLL due to COVID-19 between 2020 and 2021 at the national level, by sex, the analytical method developed by the World Health Organization, Vieira et al. y Williams et al. [12–14] was used, however, we elaborate some mathematical refinements as explained below. Formally, we define the YLL in equation:

$$YLL^{COVID-19}_{j,s,t} = \sum_{i=1}^{l} M^{COVID-19}_{j,s,t,i} * (LE_{j,s,t} - AIRP^{COVID-19}_{j,s,t,i}) \, if \, LE_{j,s,t} > AIRP^{COVID-19}_{j,s,t,i},$$

where $j$ represents a specific country, $s$ represents the sex of study (female and male), $t$ the year of reference calculation (2020, 2021), $i$ each age range of analysis, $M$ the number of deaths recorded in each age range by COVID-19 ($\epsilon R^{+}$), $LE$ life expectancy, and $AIRP$ the ages intermediate range point. If there is no disaggregation by sex, it is estimated without this differentiation by applying the same methodology.

For the segmentation by life cycle, the following division was made childhood (0–11 years), adolescence-youth (12–26 years), adulthood (27–59 years) and old age (60 years up to the life expectancy of the country $j$). When reviewing the database of deaths from COVID-19, the countries report the information based on different age groups (simple ages, five-year periods, decades, asymmetric intervals, among other forms), therefore, the following algorithm was carried out to compile the predefined groups by life cycle: i) review the age range established by the country and see if it is included in one life cycle or more; ii) if it is included in more than one, calculate the percentage of years that make up each life cycle; iii) assign the YLL, assuming a uniform probability function. For example, if country $j$ has the age range (58 years-67 years), it would be in two life cycles, adulthood and old age, then, the YLL calculated in this interval, 20% would be distributed for adulthood $\left(\frac{2 \, years}{10 \, years}; \, 57 \, and \, 58 \, years\right)$ and 80% for old age $\left(\frac{8 \, years}{10 \, years}; \, 60 \, to \, 67 \, years\right)$.

Lastly, having already calculated the YLLs per country, we estimate how much each national government should have been willing to spend from their own health budgets to manage COVID-19 (from a health system perspective, i.e. only taking into account health effects, not broader economic effects). To achieve this, the following steps were systematically performed: i) the YLLs calculated by country were multiplied by the quality of life weights used in Claxton et al. [18] (assuming the same utilities of the United Kingdom for the different countries of analysis), according to the life cycle and sex; ii) the upper limit of the estimated range for cost-effectiveness thresholds (CET) per country was taken (as a conservative scenario), developed by Woods et al. [19], and was updated to 2021 prices using national consumer price indices

[20]; and iii) the resulting in i) and ii) was multiplied, thus finding the approximate monetary value that should have been allocated in each country for the attention of the pandemic between the years 2020 and 2021.

All the statistical processes were done in the R software, 4.2.1 version (used libraries: tidyverse, openxlsx, readxl, lubridate, osfr, remotes, covidAgeData, ggplot2, countrycode, sf, latex2exp, extrafont and wbstats). Specific R scripts employed for the analysis, death calculations by country, gender, and age group, and the creation of primary and supplementary graphs are provided in the supplementary materials as S1, S2 and S3 Codes, respectively. The map illustrating the distribution of YLL (Years of Life Lost) was created using Tableau 2021.2 software, with a base map layer sourced from OpenStreetMap, an open data platform. On the other hand, this study follows the STROBE Statement checklist [21].

### Ethics statement

This is a secondary data analysis of open public sources of country level data. Therefore, our study did not require the approval of an Ethics Committee. All databases are secondary sources, published on the web.

## Results

When calculating the YLL between January 1, 2020, and December 31, 2021 (considered the deepest period of the pandemic), for the 49 countries mentioned studied, which comply with complete health information systems regarding COVID-19, we found that 85,649,579 years of life have been lost to COVID-19, for an overall rate of 398.9 YLL per 10,000 inhabitants. Among the studied countries, Peru, Hungary Czechia, Colombia and Mexico presented the highest YLL rate for this communicable disease, while Somalia, New Zealand, Togo, Australia and Haiti showed the lowest rates. As shown in Fig 1 most African countries do not report mortality data or if they do, these are of poor quality, an action unfortunately correlated with their poor epidemiological surveillance systems [22,23]. For all the deaths across the 36 study countries for which sex-disaggregated data is available, Fig 2 shows a significant rate of years of life lost for the old age life cycle, followed by the adult stage.

Although for almost all the YLL countries the distribution structure is similar in the four predefined life cycles (having a higher percentage of adulthood and old age, compared to childhood and adolescence-youth), the amount of YLL by people in childhood is minimal for men in Moldova and for women in Slovenia, Uruguay, and Denmark. Also, for almost all countries, YLL are distributed equally between both sexes (Fig 3).

In the supplementary material, S1 and S2 Figs detail the YLL per 10,000 inhabitants during the childhood life cycle for females and males, respectively. For countries without gender-disaggregated data during this life cycle, S3 Fig offers a combined perspective for both sexes. These figures indicate that Latin American countries are prominently affected, especially nations like Peru, Brazil, Mexico, Chile, Colombia, Paraguay, and Argentina. Remarkably, the maximum annual rate does not exceed 10 YLL per 10,000 inhabitants.

The trends observed in childhood are mirrored in the adolescent and youth life cycles, represented in S4 and S5 Figs for females and males, respectively, and S6 Fig for countries without gender-specific data. The highest rate in this age bracket does not go beyond 190 YLL per 10,000 inhabitants annually. For the adulthood phase, S7 and S8 Figs provide insights for females and males, respectively, with S9 Fig catering to countries lacking gender-disaggregated data. In these figures, countries such as Hungary, the Philippines, and the Czech Republic emerge with notably high rates. Lastly, during old age, S10 and S11 Figs represent female and male data respectively, while S12 Fig highlights the YLL for countries without gender

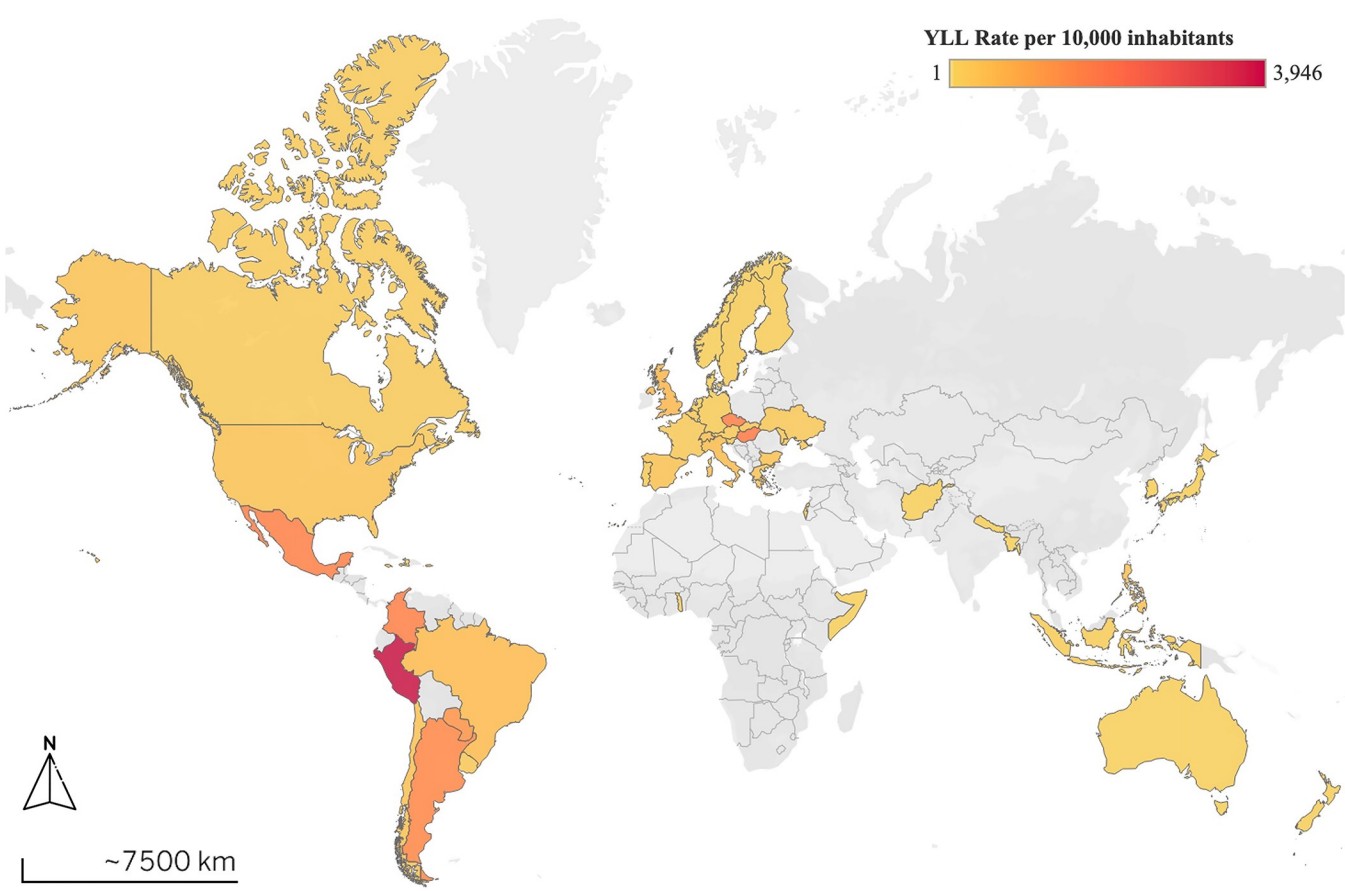

**Fig 1. Map of cumulative calculated total YLL between 2020 and 2021 per 10,000 inhabitants.** Note: the countries in gray do not have information.

distinction. In this category, several countries report more than 2,000 YLL per 10,000 female inhabitants and over 3,000 YLL per 10,000 male inhabitants on an annual basis.

For those interested, S1 Table displays the estimated rates of Years of Life Lost (YLL) per 10,000 inhabitants, broken down by gender and age group, for countries that provide gender-specific data. Meanwhile, S2 Table presents the total YLLs by gender, and when disaggregated data is not available, it combines both genders, categorized by age group for all countries with available information.

Finally, Table 1 shows the budget (in millions of purchasing power parity-adjusted US dollars, at 2021 prices) that each country should have invested in managing the COVID-19 pandemic (in decent order according to their monetary value). This could give a kind of measure of how much should be spent in the future to prevent future pandemics (for example, investments in public health preparedness, epidemiological surveillance, hospital infrastructure, etc.). As expected, countries with high YLL (e.g. Brazil, Peru) or with high CET (e.g. Switzerland, Canada), or high YLL and CET at the same time (e.g. USA, England), present the highest cost-effective national budgets. The first five places are countries of the American continent, while the last two countries belong to Africa.

## Discussion

Between January 2020 and December 2021, we found that over 85,649,579 years of life have been lost to COVID-19 among the 49 selected countries, among which 2,724,463 deaths were

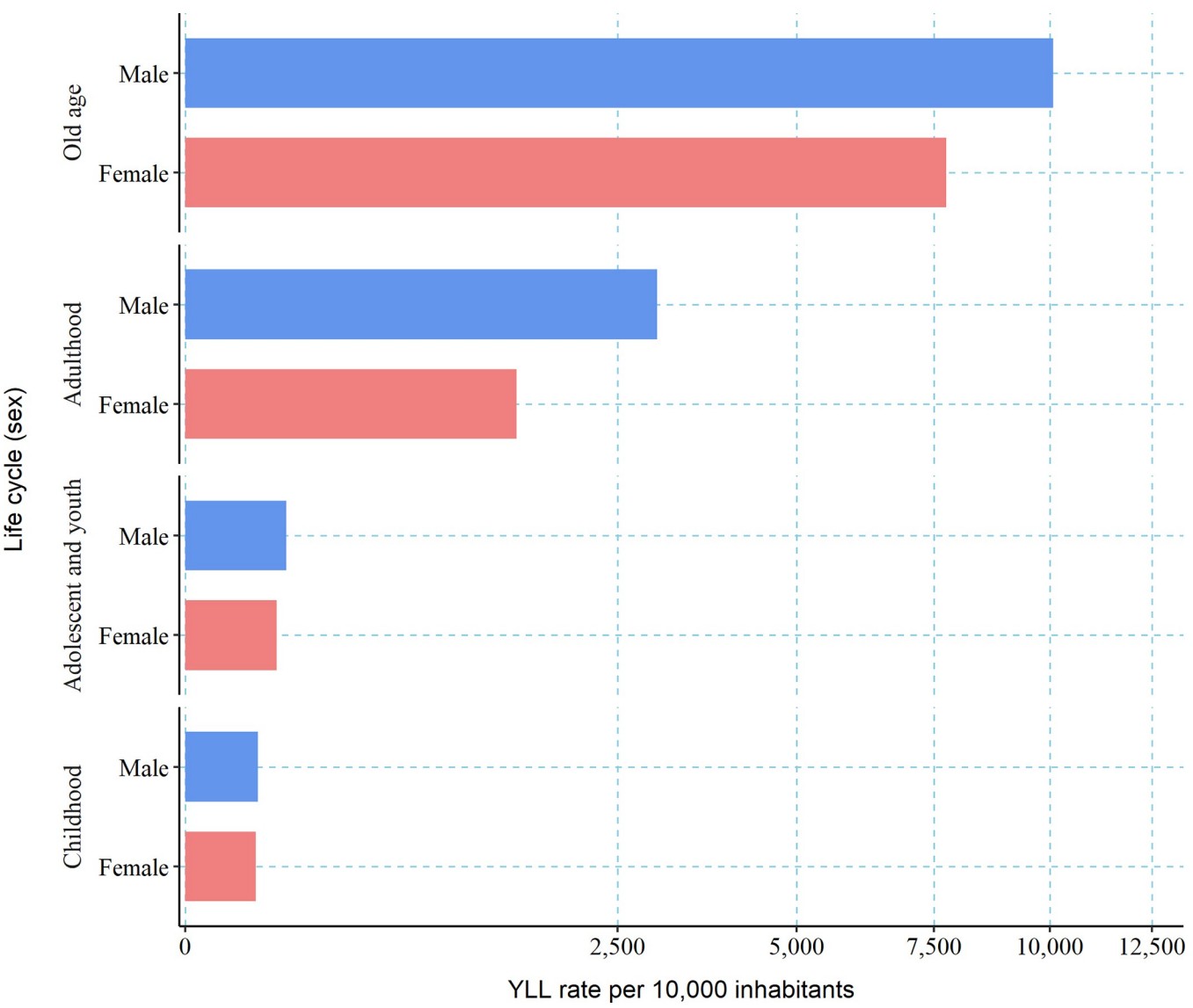

**Fig 2. YLL rate per 10,000 inhabitants, by sex and life cycle, in 2020 and 2021 in all the studied countries with available data by sex (n = 36).**

caused by the disease. While most of the YLL occurred on the American continent, after adjusting for population two central European countries were found to be in the top five with the highest rate of YLL (Hungary and Czechia). Our study offers two key results worth discussing: i. we did not find systematic differences in YLL rates by gender, but we did find differences by life cycle, and ii. health information systems must be strengthened in order to have reliable data that enables producing evidence-based and actionable insights for decision-making in public health when facing future pandemics.

Despite several studies have reported that mortality from COVID-19 is higher for men due to differences in the innate and adaptive immune system [4], but also due to higher prevalences of smoking and other risk factors for several non-communicable diseases [24], we found in our analysis that the YLL was similar for both sexes, with the exceptions of Italy, Israel, Norway and Japan, where men had higher YLL in 2021. These results diverge from previously published multi-country studies, where men carried a proportionally higher burden of

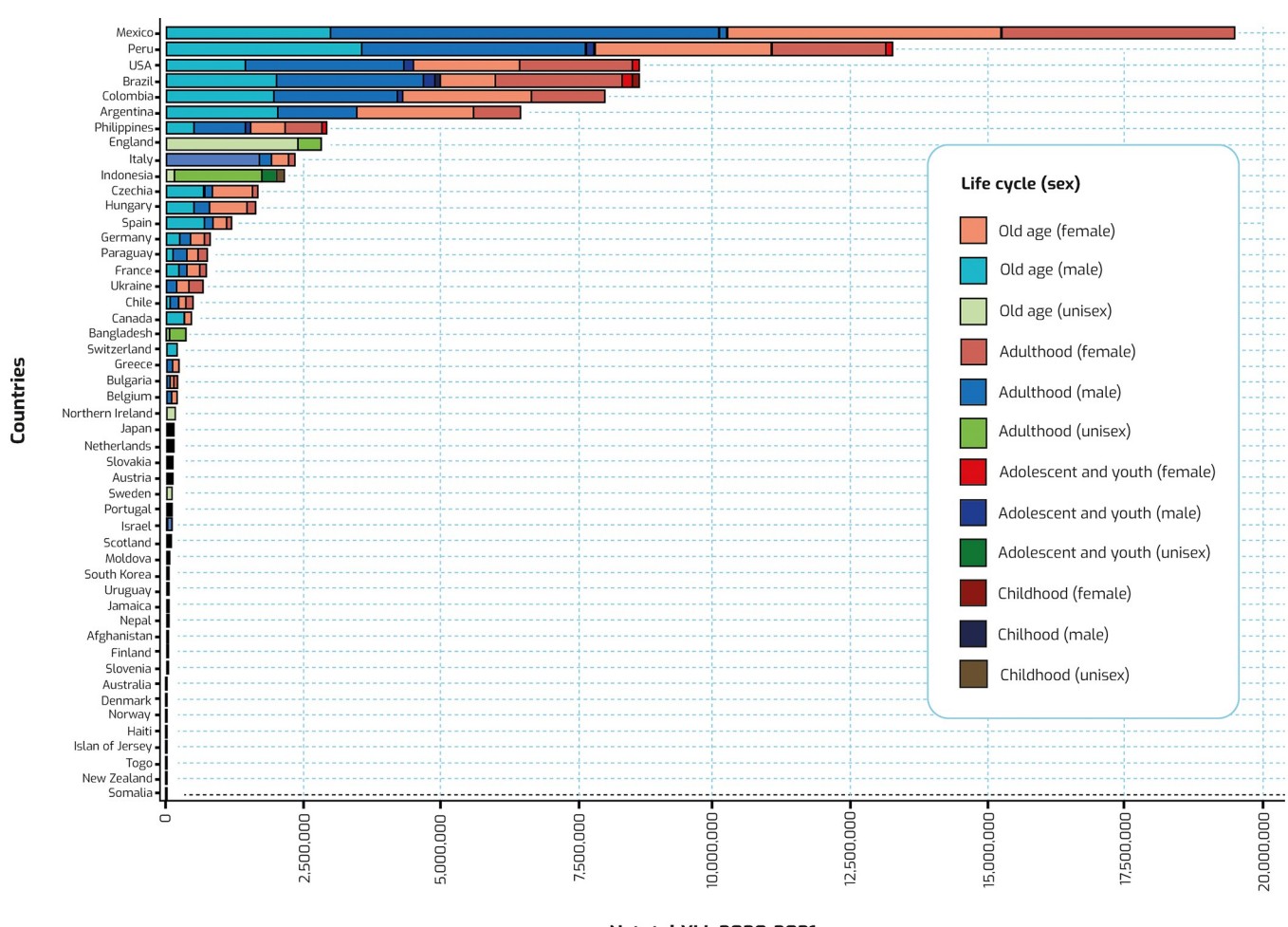

**Fig 3. Cumulative calculated total YLL, at the national level, by sex and life cycle, in 2020 and 2021.**

YLL [11,25]. Although the results could seem unexpected, since most deaths from COVID-19 occurred in men (54.4%) and they tended to die younger than women on average, this could be explained by differences in life expectancy between the two sexes. However, some countries lacked data disaggregated by sex, which limited our analysis, especially by precluding the analysis of standardized YLL rates which is appropriate.

Furthermore, we found that adults and older people had the highest burden of YLL to COVID-19. From the beginning of the pandemic, it was clear that the risk of severe illness, hospitalization, and death from COVID-19 increased with age [26]. According to US CDC estimates, the risk of death tended to increase in adulthood, so people between 30 and 39 years had a mortality rate from COVID-19 that was four times higher than people between 18 and 29 years of age. And this risk was 340 times higher among adults 85 years and over [27]. However, since the YLL indicator not only considers the number of deaths but also the ages at which they occur, then the younger the age at the time of death, the greater the number of potential years lost. In this study, we observed that deaths from COVID-19 in young people were few (globally, 2.96% of all YLL have been attributed to deaths of individuals between 0 and 26 years), and even when they would be expected to contribute the most YLL at the individual level, it was the older adults who had both the highest absolute number of deaths and YLL combined (50.01%).

**Table 1. Cost-effective budget for the national management of the COVID-19 pandemic, 2020–2021 (in millions of purchasing power parity-adjusted US dollars, at 2021 prices).**

| Country | Cost-effective budget at the national level (using QALY) | | |
| --- | --- | --- | --- |
| | **2020** | **2021** | **Total 2020–2021** |
| USA | 85,082.3 | 251,678.2 | 336,760.5 |
| Mexico | 51,626.6 | 99,744.9 | 151,371.4 |
| Brazil | 29,704.5 | 56,412.3 | 86,116.8 |
| Peru | 19,702.4 | 40,613.1 | 60,315.5 |
| Argentina* | 14,339.2 | 41,887.7 | 56,226.9 |
| England | 13,620.5 | 37,967.0 | 51,587.5 |
| Colombia | 12,481.7 | 37,091.9 | 49,573.7 |
| Italy | 5,776.0 | 29,673.8 | 35,449.9 |
| Germany | 5,265.6 | 14,794.2 | 20,059.9 |
| Czech Republic | 3,846.3 | 12,360.1 | 16,206.4 |
| Spain | 3,406.1 | 12,746.3 | 16,152.4 |
| France | 3,982.8 | 9,529.6 | 13,512.4 |
| Switzerland | 10,899.9 | 2,597.8 | 13,497.7 |
| Canada | 1,013.5 | 12,119.6 | 13,133.1 |
| Hungary | 2,270.2 | 9,908.3 | 12,178.5 |
| Chile | 1,468.0 | 3,611.5 | 5,079.6 |
| Indonesia | 902.1 | 3,567.1 | 4,469.2 |
| Belgium | 1,545.5 | 2,896.2 | 4,441.7 |
| Philippines | 872.2 | 3,462.7 | 4,334.9 |
| Netherlands | 1,153.4 | 2,229.8 | 3,383.2 |
| Sweden | 1,077.1 | 2,251.7 | 3,328.8 |
| Austria | 801.5 | 2,265.3 | 3,066.8 |
| Ukraine | 600.1 | 2,453.6 | 3,053.7 |
| Northern Ireland | 706.2 | 2,160.2 | 2,866.4 |
| Japan | 283.1 | 1,949.6 | 2,232.7 |
| Greece | 414.2 | 1,811.4 | 2,225.7 |
| Paraguay | 231.2 | 1,610.5 | 1,841.7 |
| Israel | 534.5 | 1,045.9 | 1,580.4 |
| Scotland | 433.6 | 1,132.0 | 1,565.6 |
| Portugal | 259.4 | 778.0 | 1,037.5 |
| Slovakia | 219.6 | 805.5 | 1,025.1 |
| Bulgaria | 235.6 | 681.1 | 916.7 |
| Norway | 217.6 | 678.4 | 896.0 |
| Australia | 121.9 | 647.2 | 769.0 |
| Finland | 585.3 | 180.2 | 765.5 |
| Uruguay | 9.9 | 706.1 | 716.0 |
| Denmark | 145.0 | 417.8 | 562.8 |
| South Korea | 69.7 | 492.4 | 562.2 |
| Slovenia | 93.9 | 265.3 | 359.3 |
| Island of Jersey* | 93.6 | 144.4 | 238.0 |
| Bangladesh | 81.8 | 125.9 | 207.7 |
| Jamaica | 8.8 | 125.4 | 134.2 |
| Moldova | 27.6 | 57.3 | 84.9 |
| New Zealand | 15.9 | 14.6 | 30.5 |
| Nepal | 9.9 | 8.9 | 18.8 |

*(Continued)*

**Table 1.** (Continued)

| Country | Cost-effective budget at the national level (using QALY) | | |
| --- | --- | --- | --- |
| | **2020** | **2021** | **Total 2020–2021** |
| Afghanistan | 4.8 | 8.9 | 13.7 |
| Haiti | 2.6 | 6.0 | 8.6 |
| Togo | 0.2 | 0.7 | 0.9 |
| Somalia* | 0.0 | 0.3 | 0.3 |

Note: * countries in which the CET was taken as one GDP per capita.

This pattern is consistent with the results reported by Pifarré et al. who found a higher burden of YLL among adults older than 55 years–nearly 70% of the 20.5 million YLL to COVID-19 in the 81 countries that they analyzed during the first year of the pandemic [11]–. Also, Williams et al. reported a higher burden of YLL in older populations in the 20 countries that they studied [13]. It is worth noting that while we observed a higher proportion of YLL in childhood, youth and adolescence in African countries, after adjusting for population size the highest rates among young people were in countries in the Americas. Thus, in some countries such as Brazil, USA, Indonesia, Mexico, Peru y Colombia the younger population had a higher burden of YLL.

According to our results, upper-middle and high-income countries presented higher rates of YLL. This finding contrasts with what is reported in the literature on low-income countries having a higher mortality burden from COVID-19 [28], but also mortality excess which also consider indirect deaths. According to Oxfam, even though wealthy countries have also experienced devastating effects due to the pandemic, low-income countries have been the most affected not only in terms of the magnitude of specific mortality–deaths have been up to four times higher than in wealthy nations [29]–, but also in other direct and indirect impacts from this syndemic phenomenon [30], as a social, economic, ecological and health crisis [31].

However, this finding also suggests a potential angular matter: the importance of having strengthened information systems to effectively address a syndemic like COVID-19. We have already mentioned how the capacity and performance of the countries' national epidemiological surveillance systems, their capacity for testing and contact tracing, effective access to health services, and the quality and coverage of vital statistics systems can affect evidence-based actions, morbidity and mortality outcomes and precision when evaluating the impacts of the pandemic. Thus, it is possible that the higher rates of YLL observed in upper-middle- and high-income countries are due to their being more able to record events than lower-income countries, rather than to higher premature mortality. In this regard, 12 countries stand out with information systems that had the best degree of disaggregation of mortality from COVID-19 by sex and age: Argentina, Chile, Colombia, Czechia, Hungary, Mexico, Moldova, Paraguay, Peru, Philippines and the USA. In fact, some of these countries were the ones that were ranked with the highest rates of YLL in our study.

This limitation of the study is very important when using YLL, since suspicious deaths are not considered in the first place, neither underreported deaths from COVID-19 were included. Additionally, from a broader perspective, our analysis is unable to identify indirect deaths, given by the COVID-19 pandemic as a social crisis that impacted health systems, and society as a whole in all countries. A broader consideration is likely to show worse results in low-income countries, but this would require a different methodological approach. Therefore, the ranks of the identified countries should be read with the consideration that these also reflect the quality of the registry, and that the YLL also increases due to the capacity of the surveillance

systems and the quality of the registry, and not always, represents actual deaths. For example, in other previous comparisons it has been found that countries like Colombia showed a specific mortality rate that converges with the excess mortality rate, in contrast to other countries like India that did not participate in this study [32]. In addition, according to Wollburg et al., the pandemic caused heterogeneous disruptions in National Statistics Offices, not only increasing the demand for data but also altering their ability to produce statistics. In this regard, the operations have been significantly affected in low- and lower-middle-income countries. Thus, pre-existing inequalities among countries in the quality and availability of information have worsened due to the pandemic [33].

In addition, a limitation of our study with respect to estimating the cost-effective budget for national COVID-19 management relates to the extrapolation of the United Kingdom quality of life weights to all other countries in the study. Although the ideal is to use the specific values of each country, almost none of these have them calculated. Obtaining them presents psychometrics and clinimetrics challenges that do not yet have a comprehensive solution and are still being studied by health economists [34,35].

On the other hand, the COVID-19 pandemic required sufficient national funding to ensure a comprehensive response. However, strong budgetary restrictions and limited power of action were contemplated in different health systems, especially in low- and middle-income countries, which affected the development of effective detection, tracking, and isolation programs, the provision of ICU beds and highly trained clinical teams, as well as the purchase of timely vaccines against COVID-19. When estimating the cost-effective budget to address the pandemic between 2020 and 2021, the limitations of activities and resources that a good proportion of countries have to face in future pandemics are evident. This should draw attention to support informed and evidence-based policies that manage to optimize the little budgetary resource that is available.

One of the notable strengths of our study is the observation time. To the best of our knowledge, ours is the first study to analyze a long critical period during the COVID-19 pandemic in multiple countries, the years 2020 and 2021, which were the first two years of the pandemic and when the highest peaks in virus transmission occurred. Other studies have analyzed shorter periods, primarily the first year of the pandemic [12,25,36].

Our study also has other limitations. It was limited to the 49 countries that had quality data and disaggregation by age of mortality from COVID-19, which constitutes approximately 25% of the countries in the world. We were unable to include countries with large populations such as China, India, and Pakistan. Therefore, extrapolation to countries not analyzed should be avoided because they could be different in terms of crucial aspects of their responses to the pandemic, and this could lead to different results than those observed here. It is also worth mentioning that the COVID-19 pandemic and the measures to address it could have caused indirect effects that cannot be observed in our analysis. Lastly, the accuracy of our results depends on the quality of the national mortality databases, mainly from the registry of deaths caused by COVID-19.

## Conclusion

Our study provides evidence of the high impact of the COVID-19 pandemic in terms of years of lives lost, which goes further than studies of specific mortality. We have found differentiated impacts of the pandemic among countries, which may be related not only to their structural and pre-existing conditions, but also to national responses, policies, and actions to overcome the crisis. The most affected countries need to be supported, considering those with a significant burden of YLL among young people. Lastly, concentrating efforts on strengthening

national health information systems should be an immediate priority, especially in those countries without available data.

## Supporting information

**S1 Fig. YLL (Years of Life Lost) per 10,000 female inhabitants during childhood by country, 2020–2021.**
(TIFF)

**S2 Fig. YLL (Years of Life Lost) per 10,000 male inhabitants during childhood by country, 2020–2021.**
(TIFF)

**S3 Fig. YLL (Years of Life Lost) per 10,000 inhabitants (both sexes) during childhood by country, 2020–2021 -in countries without gender-disaggregated data.**
(TIFF)

**S4 Fig. YLL (Years of Life Lost) per 10,000 Female inhabitants during adolescence and youth by country, 2020–2021.**
(TIFF)

**S5 Fig. YLL (Years of Life Lost) per 10,000 male inhabitants during adolescence and youth by country, 2020–2021.**
(TIFF)

**S6 Fig. YLL (Years of Life Lost) per 10,000 inhabitants (both sexes) during adolescence and youth by country, 2020–2021 -in countries without gender-disaggregated data.**
(TIFF)

**S7 Fig. YLL (Years of Life Lost) per 10,000 female inhabitants during adulthood by country, 2020–2021.**
(TIFF)

**S8 Fig. YLL (Years of Life Lost) per 10,000 male inhabitants during adulthood by country, 2020–2021.**
(TIFF)

**S9 Fig. YLL (Years of Life Lost) per 10,000 inhabitants (both sexes) during adulthood by country, 2020–2021 -in countries without gender-disaggregated data.**
(TIFF)

**S10 Fig. YLL (Years of Life Lost) per 10,000 female inhabitants during old age by country, 2020–2021.**
(TIFF)

**S11 Fig. YLL (Years of Life Lost) per 10,000 male inhabitants during old age by country, 2020–2021.**
(TIFF)

**S12 Fig. YLL (Years of Life Lost) per 10,000 inhabitants (both sexes) during old age by country, 2020–2021 -in countries without gender-disaggregated data.**
(TIFF)

**S1 Table. Accumulative Years of Life Lost (YLL) rates per 10,000 inhabitants by age group, sex, and country: 2020–2021.**
(XLSX)

**S2 Table. Accumulative Years of Life Lost (YLL) by age group, sex, and country: 2020–2021.**
(XLSX)

**S1 Code. R Code for analysis conducted in the study.**
(R)

**S2 Code. R Code procedure for calculating deaths by country, gender, and age group.**
(R)

**S3 Code. R Code for generating main and supplementary material graphs.**
(R)

## Acknowledgments

Oscar Espinosa would like to express his appreciation to Paul Revill and Ilias Kyriopoulos, for their valuable and constructive suggestions regarding this research.

## Author Contributions

**Conceptualization:** Oscar Espinosa.

**Data curation:** Oscar Espinosa, Jeferson Ramos.

**Formal analysis:** Oscar Espinosa, Jeferson Ramos, Maylen Liseth Rojas-Botero.

**Investigation:** Oscar Espinosa, Jeferson Ramos, Maylen Liseth Rojas-Botero, Julián Alfredo Fernández-Niño.

**Methodology:** Oscar Espinosa, Jeferson Ramos, Maylen Liseth Rojas-Botero.

**Project administration:** Oscar Espinosa.

**Supervision:** Oscar Espinosa.

**Validation:** Oscar Espinosa, Jeferson Ramos, Julián Alfredo Fernández-Niño.

**Visualization:** Oscar Espinosa, Jeferson Ramos, Maylen Liseth Rojas-Botero.

**Writing – original draft:** Oscar Espinosa, Jeferson Ramos, Maylen Liseth Rojas-Botero, Julián Alfredo Fernández-Niño.

**Writing – review & editing:** Oscar Espinosa, Jeferson Ramos, Maylen Liseth Rojas-Botero, Julián Alfredo Fernández-Niño.

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
