## [Decision Letter · Decision Letter 0]

29 May 2023

PGPH-D-23-00567

Years of life lost to COVID-19 in 49 countries: A gender- and life cycle-based analysis of the first two years of the pandemic

Dear Dr. Julian Alfredo Fernandez-Nino,

Thank you for submitting your manuscript to PLOS Global Public Health. After careful consideration, we feel that it has merit but does not fully meet PLOS Global Public Health’s publication criteria as it currently stands. Therefore, we invite you to submit a revised version of the manuscript that addresses the points raised during the review process.

We look forward to receiving your revised manuscript.

Kind regards,

Thu-Anh Nguyen

Academic Editor

Journal Requirements:

1. Please provide a copy edited of your paper.

2. Some material included in your submission may be copyrighted. According to PLOS’s copyright policy, authors who use figures or other material (e.g., graphics, clipart, maps) from another author or copyright holder must demonstrate or obtain permission to publish this material under the Creative Commons Attribution 4.0 International (CC BY 4.0) License used by PLOS journals. Please closely review the details of PLOS’s copyright requirements here: PLOS Licenses and Copyright. If you need to request permissions from a copyright holder, you may use PLOS's Copyright Content Permission form.

Potential Copyright Issues:

Figure 1: please (a) provide a direct link to the base layer of the map (i.e., the country or region border shape) and ensure this is also included in the figure legend; and (b) provide a link to the terms of use / license information for the base layer image or shapefile. We cannot publish proprietary or copyrighted maps (e.g. Google Maps, Mapquest) and the terms of use for your map base layer must be compatible with our CC-BY 4.0 license. 

Additional Editor Comments (if provided):

Reviewers' comments:

Reviewer's Responses to Questions

**Comments to the Author**

1. Does this manuscript meet PLOS Global Public Health’s publication criteria? Is the manuscript technically sound, and do the data support the conclusions? The manuscript must describe methodologically and ethically rigorous research with conclusions that are appropriately drawn based on the data presented.

Reviewer #1: Yes

Reviewer #2: Yes

2. Has the statistical analysis been performed appropriately and rigorously?

Reviewer #1: Yes

Reviewer #2: Yes

3. Have the authors made all data underlying the findings in their manuscript fully available (please refer to the Data Availability Statement at the start of the manuscript PDF file)?

Reviewer #1: Yes

Reviewer #2: Yes

4. Is the manuscript presented in an intelligible fashion and written in standard English?

Reviewer #1: Yes

Reviewer #2: Yes

5. Review Comments to the Author

Reviewer #1: Comments

General comments

This is an interesting and insightful paper. The value of using YLL rather than deaths as a measure of pandemic ‘performance’, in the context of COVID-19, is well described and presented. I think that this paper is worthy of publication. However, there are some issues that should first be addressed, as detailed below.

Specific areas to address

Figures/Tables:

1. Figure 1 does not currently make sense with ‘Unisex’ presented alongside ‘Male’ and ‘Female’. If only looking at the axis and not the values, it may be assumed that ‘Unisex’ would be showing a combined total for the two sexes presented. I do not think that this Figure should be presented as is, without country-specific information that makes clear why the ‘Unisex’ values are so different to the ‘Male’ and ‘Female’ values. The Supplementary Figures provide more useful information, in separating countries that do/don’t have data by sex clearly. A more meaningful ‘Figure 1’ could be to present an age-standardised YLL rate for all countries (separating countries by sex where available, as currently done in the supplementary figures), or show an aggregate of sex, for each life cycle stage (i.e. grouping all countries).

2. Please state the currency of budgets presented in Table 1.

Methods:

3. The economic analysis is an interesting additional assessment of country-specific performance, and potential estimate for future pandemic planning. However, there are limitations that have not been addressed. Utility weights from the UK have been applied (from the EQ-5D, as stated in the cited study, reference 18: Claxton et al., though this should be stated more clearly in the present paper also), but these are highly unlikely to be accurate for all 49 countries. Secondly, health valuation has been found to vary greatly between populations when using the EQ-5D utility measure (see for discussions: Gerlinger, C., Bamber, L., Leverkus, F. et al. Comparing the EQ-5D-5L utility index based on value sets of different countries: impact on the interpretation of clinical study results. BMC Res Notes 12, 18 (2019); and: Roudijk B, Donders ART, Stalmeier PFM; Cultural Values Group. Cultural Values: Can They Explain Differences in Health Utilities between Countries? Med Decis Making. 2019 Jul;39(5):605-616). This limitation, and the lack of cross-walking or use of country-specific values for quality of life should be clearly addressed in the discussion.

4. Please state whether discounting as been applied to YLL calculations (it is inferred that YLLs are not discounted in this paper, but this should be stated/justified). Note that UK recommendations are to discount costs and benefits at 3.5% per year (Claxton et al.) Presenting discounted results in the supplement would be more thorough and provide this option for readers.

Results:

5. Two statements are made regarding differences by sex, which are currently conflicting. On page 9, line 156-157, it is stated that a higher burden was found for males by all life cycle stages. However, on page 13, line 207, it is stated that no systematic differences in YLLs were found by gender. Please clarify these statements.

Discussion

6. The main limitations of the study are well addressed in the discussion, particularly with regard to likely underreporting of COVID-19 deaths in lower income countries due to poor health information system capacity/strength. However, given this limitation, have the authors considered calculating an uncertainty range for each country’s YLL estimate, perhaps with a measure of health information system strength being used to assign different levels of uncertainty in presented YLL? If this is not possible, the authors should consider presenting an additional sub-analysis, grouping countries by a proxy of health information system strength (e.g. low/mid/high income countries), to avoid inaccurate comparisons being made between countries with likely high variation in the accuracy of reported deaths.

Reviewer #2: The authors have performed excellent work in this study.

Abstract:

The abstract should provide a more detailed explanation of the population, sampling, procedure of the Years of Life Lost (YLL), and statistical methods used rather than merely stating the aim of the study. Additionally, the duration of the study should not be repeated in the abstract. Instead of saying, "We did not detect differences," it would be more appropriate to state "We found no difference."

Main Text:

It is recommended to adhere to the reporting guidelines outlined in the Strengthening the Reporting of Observational Studies in Epidemiology (STROBE) statement. Following these guidelines will enhance the reporting of the analysis section and the measurement section of YLL and QALY. Moreover, it is important to provide recommendations regarding the policy-level implications of this study in the current stage of the pandemic or for future pandemics.

The results should be reported under appropriate subheadings, which will reflect the research question raised in the introduction of the study.

6. PLOS authors have the option to publish the peer review history of their article (what does this mean?). If published, this will include your full peer review and any attached files.

**Do you want your identity to be public for this peer review?** For information about this choice, including consent withdrawal, please see our Privacy Policy.

Reviewer #1: No

Reviewer #2: **Yes: **Humayun Kabir

---

## [Decision Letter · Decision Letter 1]

13 Aug 2023

PGPH-D-23-00567R1

Years of life lost to COVID-19 in 49 countries: A gender- and life cycle-based analysis of the first two years of the pandemic

Dear Dr. Fernández-Niño,

Thank you for submitting your manuscript to PLOS Global Public Health. After careful consideration, we feel that it has merit but does not fully meet PLOS Global Public Health’s publication criteria as it currently stands. Therefore, we invite you to submit a revised version of the manuscript that addresses the points raised during the review process.

We look forward to receiving your revised manuscript.

Kind regards,

Jianhong Zhou

Staff Editor

Journal Requirements:

2. We do not publish any copyright or trademark symbols that usually accompany proprietary names, eg (R), (C), or TM  (e.g. next to drug or reagent names). Please remove all instances of trademark/copyright symbols throughout the text, including ® on pages 6.

Additional Editor Comments (if provided):

Reviewers' comments:

Reviewer's Responses to Questions

**Comments to the Author**

1. If the authors have adequately addressed your comments raised in a previous round of review and you feel that this manuscript is now acceptable for publication, you may indicate that here to bypass the “Comments to the Author” section, enter your conflict of interest statement in the “Confidential to Editor” section, and submit your "Accept" recommendation.

Reviewer #1: (No Response)

Reviewer #2: All comments have been addressed

2. Does this manuscript meet PLOS Global Public Health’s publication criteria? Is the manuscript technically sound, and do the data support the conclusions? The manuscript must describe methodologically and ethically rigorous research with conclusions that are appropriately drawn based on the data presented.

Reviewer #1: Yes

Reviewer #2: (No Response)

3. Has the statistical analysis been performed appropriately and rigorously?

Reviewer #1: Yes

Reviewer #2: (No Response)

4. Have the authors made all data underlying the findings in their manuscript fully available (please refer to the Data Availability Statement at the start of the manuscript PDF file)?

Reviewer #1: Yes

Reviewer #2: (No Response)

5. Is the manuscript presented in an intelligible fashion and written in standard English?

Reviewer #1: Yes

Reviewer #2: (No Response)

6. Review Comments to the Author

Reviewer #1: Thank you to the authors for addressing my comments. I believe this work is suitable for publication (given some final comments below).

1. In relation to my original question regarding currency: the currency is shown in the table heading and in the text as 'dollars'. This should be more clearly specified, as done in the referenced study by Woods et al., which notes that CETs are presented in purchasing power parity adjusted US dollars (USD).

2. On page 10, line 225: the authors state that deaths and YLLs combined for older adults adds to 947.0%. There is also a closed bracket after this number, with no open bracket.

3. Figure 2 now makes more sense. However, in the text (page 6 line 153-155) the authors state that this figure shows data from all 49 countries (with the figure title also stating 'all studies countries'). This figure presents results by sex, so would seemingly only include the countries that had data available by sex (36 countries, per the Methods section). Please update the text and figure title as needed.

Reviewer #2: No comments at this stage of revision.

7. PLOS authors have the option to publish the peer review history of their article (what does this mean?). If published, this will include your full peer review and any attached files.

**Do you want your identity to be public for this peer review?** For information about this choice, including consent withdrawal, please see our Privacy Policy.

Reviewer #1: No

Reviewer #2: No

---

## [Editor Report · Decision Letter 2]

24 Aug 2023

Years of life lost to COVID-19 in 49 countries: A gender- and life cycle-based analysis of the first two years of the pandemic

PGPH-D-23-00567R2

Dear Dr. Fernández-Niño,

We are pleased to inform you that your manuscript 'Years of life lost to COVID-19 in 49 countries: A gender- and life cycle-based analysis of the first two years of the pandemic' has been provisionally accepted for publication in PLOS Global Public Health.

Best regards,

Julia Robinson

Executive Editor